# *gp2Scale*: A Class of Compactly Supported Non-Stationary Kernels and Distributed Computing for Exact Gaussian Processes on 10 Million Data Points

**Marcus M. Noack** [* 1] **Mark D. Risser** [2] **Hengrui Luo** [3 1] **Vardaan Tekriwal** [1 4] **Ronald J. Pandolfi** [1]

## Abstract

Despite a large corpus of recent work on scaling up Gaussian processes, a stubborn trade-off between computational speed, prediction, and uncertainty quantification accuracy, and customizability persists. This is because the vast majority of existing methodologies exploit various levels of approximation that lower accuracy and limit the flexibility of kernel and noise-model designs — an unacceptable drawback at a time when expressive non-stationary kernels are on the rise in many fields. Here, we propose a methodology we term *gp2Scale* that scales exact Gaussian processes to more than 10 million data points without relying on inducing points, kernel interpolation, or neighborhood-based approximations, and instead leverages the existing capabilities of a GP: its kernel design. Highly flexible, compactly supported, and non-stationary kernels lead to the identification of naturally occurring sparse structure in the covariance matrix, which is then exploited to compute the linear system solution and the log-determinant for training. We demonstrate our method's functionality on several real-world datasets and compare it with state-of-the-art approximation algorithms. Although we show superior approximation performance in many cases, the method's real power lies in its agnosticism toward arbitrary GP customizations — core kernel design, noise, and mean functions — and the type of input space, making it optimally suited for modern Gaussian process applications.

[1] Applied Mathematics and Computational Research Division, Lawrence Berkeley National Laboratory, 1 Cyclotron Rd, Berkeley, CA 94720 [2] Climate and Ecosystem Sciences Division, Lawrence Berkeley National Laboratory, 1 Cyclotron Rd, Berkeley, CA 94720 [3] Department of Statistics, Rice University, Houston, TX, 77005 [4] UC Berkeley, 1 Sproul Hall, Berkeley, CA 94720. Correspondence to: Marcus M. Noack <MarcusNoack@lbl.gov>.

*Proceedings of the $43^{rd}$ International Conference on Machine Learning*, Seoul, South Korea. PMLR 306, 2026. Copyright 2026 by the author(s).

## 1. Introduction

Gaussian process (GP) regression is a general-purpose tool for stochastic function approximation from data. A GP is characterized by a prior normal distribution $p(\mathbf{f})$ over function values $\mathbf{f} = [f(x_1), f(x_2), ..., f(x_{|\mathcal{D}|})]^T$ that is defined by a mean $\mathbf{m} = [m(x_1), ..., m(x_{|\mathcal{D}|})]^T$, we assume zero mean without loss of generality, and a covariance $\mathbf{K} = Cov(\mathbf{f}, \mathbf{f}), \mathbf{K} \in \mathbb{R}^{|\mathcal{D}| \times |\mathcal{D}|}$. The function $f(x)$ is considered the ground-truth and data-generating function. Data $\mathcal{D} = \{(x_i, y_i)\}_{i=1}^{|\mathcal{D}|}$ is thought of to have been generated through the functional relationship $y_i = f(x_i) + \epsilon(x_i)$, where $\epsilon(x_i)$ is normally distributed noise. We will refer to the set of all $x_i$ as $\mathbf{X}$ and the vector of all measurements $y_i$ as $\mathbf{y}$. Training a GP involves evaluating the log marginal likelihood $p(\mathbf{y})$, which requires calculating a linear system solution $\mathbf{Ka} = \mathbf{y}$, for a vector $\mathbf{a}$, and the log-determinant $\log(|\mathbf{K}|)$. Both calculations scale $\mathcal{O}(|\mathcal{D}|^3)$, which has led to the widely held belief that GPs can only be applied to moderately sized datasets of <10000 points (Williams & Rasmussen, 2006). In addition, storing the covariance matrix scales at $\mathcal{O}(|\mathcal{D}|^2)$, which is often even more problematic than the time scaling because it imposes a hard limit on a given computing architecture.

The fundamental problem of scaling Gaussian processes (GPs) stems from the widely held view that the covariance matrix is inherently dense. Sparsity is entirely derived from the chosen kernel function; even if many function values $\mathbf{f}$ are uncorrelated ($Cov(f_i, f_j) = 0$), most kernel functions would be unable to assign a zero covariance, rendering the covariance matrix dense by construction. Existing kernels that are compactly supported, and therefore able to return zero, can do so only in a stationary, purely distance-related manner, which limits accuracy; this is often referred to as covariance tapering (Furrer et al., 2006; Kaufman et al., 2008). The missing scalability of Gaussian processes comes down to a model misspecification problem; providing kernels with non-stationary compact support allows the GP to discover naturally occurring sparsity in the dataset; sparse linear algebra then leads to more favorable scaling — a methodology we call *gp2Scale*.

This basic principle has been applied before by (Noack et al., 2023) to demonstrate a proof-of-concept run on a 5-million-point climate dataset. However, that early work focused on a single kernel design, did not compare with other scalable GP methodologies, and did not provide a comprehensive software framework. In this paper, we officially announce *gp2Scale* as a methodology and software to scale exact GPs to millions of data points. We also extend the methodology by (Noack et al., 2023) by defining a new class of compactly supported non-stationary kernels and performing rigorous comparisons to several state-of-the-art approximation methods. To set expectations: we do not expect that our exact GP will be as fast as some of the approximate methods, and we will require more computing power — after all, we are running an exact GP — but we will show that large-scale exact GPs are feasible and lead to competitive or better accuracy while preserving the natural flexibility of a standard GP.

*gp2Scale*, in a nutshell, has three components: (1) A flexible non-stationary compactly supported kernel function that allows natural sparsity to be discovered — not induced like in most approximate methods, (2) a distributed-computation framework that allows quick computation of the covariance matrix, and (3) a customized block-Metropolis-Hastings Markov-chain Monte Carlo (BMH-MCMC) that allows quick convergence, natural regularization, and user-friendliness.

## 2. Related Work

Past work in this field can be divided into two branches: *exact* and *approximate* methods. This distinction is driven by whether the full dataset and the associated covariances are considered or not; of course, any numerical procedure is approximate by nature, but numerical approximations are considered to lie within the exact GP category. The most notable work on exact GPs was done in (Wang et al., 2019), where the authors scaled exact GPs to over 1 million data points by avoiding the log-determinant entirely and instead taking advantage of fast conjugate-gradient iterations on GPUs to calculate the gradient of the log marginal likelihood for a local optimization. The method may be sub-optimal for non-standard kernels with many hyperparameters because, one, the gradient for each hyperparameter has to be computed, which will consume time, and two, for those kernels, the log marginal likelihood is strongly non-convex, exhibiting many local optima that render a purely local optimization ineffective.

The vast majority of work on GP scalability has focused on finding approximate solutions. Two broad families have emerged: inducing-point methods, which introduce a smaller set of points to represent the latent function, and local approximation methods, which exploit neighborhood structures for reduced computation.

Inducing-point methods introduce $M \ll |\mathcal{D}|$ pseudo-inputs $\mathbf{Z}$ to construct a lower-rank approximation of the covariance matrix. Formally, one may approximate $\mathbf{f}$ by conditioning on $\mathbf{u} = f(\mathbf{Z})$, where $\mathbf{u} \sim \mathcal{N}(0, \mathbf{K}(\mathbf{Z}, \mathbf{Z}))$. A common approach is to exploit the relationship $\mathbf{f} \approx \mathbf{K}(\mathbf{X}, \mathbf{Z})\mathbf{K}(\mathbf{Z}, \mathbf{Z})^{-1}\mathbf{u}$. Within this framework, Sparse Variational Gaussian Processes (SVGP, (Hensman et al., 2013)) stand out for their flexibility, as they can be trained by maximizing the Evidence Lower Bound (ELBO), thereby accommodating a wide range of kernels. Sparse Gaussian Process Regression (SGPR, (Snelson & Ghahramani, 2005)) likewise employs inducing points, but optimizes the marginal likelihood directly. SVGP with Contour Integral Quadrature (SVGP-CIQ, (Pleiss, 2020)) improves matrix inversion for Matérn kernels through numerical contour integration, but hinges on this kernel class; while (Luo et al., 2022) extends the sparsification methodology to a fully Bayesian additive setting (SAGP). Scalable Kernel Interpolation (SKI, (Wilson & Nickisch, 2015; Wilson et al., 2015)) and its extension, Kernel Interpolation for Scalable Structured GPs (KISS-GP, (Wilson & Nickisch, 2015)), arrange inducing points on grids to exploit structured kernel matrices and enable efficient matrix-vector multiplications.

Local approximation strategies such as Nearest-Neighbor Gaussian Processes (NNGP, (Datta et al., 2016)), Variational Nearest Neighbor Gaussian Processes (VNNGP, (Wu et al., 2022)), and the Vecchia approximation (Vecchia, 1988; Katzfuss & Guinness, 2021) approach the covariance structure by examining local subsets of the data, thereby reducing both computational cost and memory demand. In comparison to inducing-point-style methods, local approximation strategies utilize sparse precision matrices rather than sparsity in the covariance matrix and take advantage of selective conditioning on a set of neighboring points.

Selecting the most informative subset of these methods for comparison benefits from considering factors such as scalability, flexibility, approximation accuracy, and implementation complexity. SVGP, despite its reliance on variational inference, is well-established as a general-purpose inducing-point method that gracefully handles unstructured data and multiple likelihoods. SGPR, though occasionally tighter in its regression-specific marginal likelihood optimization, offers fewer advantages in broader GP applications. SVGP-CIQ, with its reliance on Matérn kernels, does not match SVGP's broader kernel compatibility. SKI emerges as a particularly strong representative of structured interpolation because it efficiently handles moderately-sized datasets without overly restrictive assumptions, apart from its dimension. Among local approaches, NNGP remains the canonical nearest-neighbor strategy for large-scale spatial data, offering significant computational gains by localizing

predictions. VNNGP combines the inducing-point method from SVGP with sparsification of the covariance matrix, similar to NNGP. Vecchia's sequential factorization covers a wide range of spatial-data scenarios and remains tractable if the data can be sorted or grouped logically.

In light of these considerations, we compare four methods to our proposed *gp2Scale* that collectively capture the essential design principles in GP scalability: SVGP as a general variational inducing-point framework, VNNGP as a paradigmatic hybrid method, taking advantage of inducing points and a notion of locality, SKI as a structured interpolation approach that is quick across low-dimensional datasets, and Vecchia as a flexible local approximation leveraging conditional independence. These four approaches span the core strategies — variational approximations, local factorizations, and kernel interpolation — while retaining broad applicability and interpretability for large-scale Gaussian process inference.

**Contributions** In this work, we propose *gp2Scale*: a new class of non-stationary compactly supported kernels that, together with HPC distributed computing and a tailored block-MCMC, allows us to scale exact GPs to millions of data points, preserving a GP's original accuracy and flexibility. We extend the framework to non-Euclidean input domains and provide comprehensive comparisons with state-of-the-art approximation methods across mid- and large-scale benchmarks. The core premise is that the GP covariance matrix is not naturally dense, but it is destined to be due to traditional kernel designs. Giving non-stationary kernels extra flexibility and compact support will allow the training to uncover sparse structure in the data, which translates into a sparse covariance matrix $\mathbf{K}$, which in turn, leads to faster linear solves and log-determinant calculations, all while the GP stays exact and maintains all of its natural flexibility regarding noise and kernel functions. In particular, since the method is based on flexible non-stationary kernels, it is agnostic to user-defined kernel designs or abstract input spaces.

## 3. Background

We consider a Gaussian prior $p(\mathbf{f}) = \mathcal{N}(\mathbf{m}, \mathbf{K})$, where $\mathbf{m} = m(x_i) \ \forall i$ is the prior mean and $\mathbf{K} = k(x_i, x_j)$. $k(x_i, x_j)$ is the kernel or covariance function. We further consider, without loss of generality, a normal likelihood $p(\mathbf{y}|\mathbf{f}) = \mathcal{N}(\mathbf{f}, \mathbf{V})$, where $\mathbf{V}$ is some noise matrix. Following Bayes' theorem, the log marginal likelihood can be derived as

$$\log(p(\mathbf{y}|\phi)) \propto$$
$$-\frac{1}{2}(\mathbf{y} - \mathbf{m}(\phi))^T(\mathbf{K}(\phi) + \mathbf{V}(\phi))^{-1}(\mathbf{y} - \mathbf{m}(\phi))$$
$$-\frac{1}{2}\ln(|\mathbf{K}(\phi) + \mathbf{V}(\phi)|), \tag{1}$$

where $\phi$ is a set of hyperparameters. Going forward, we can ignore the prior mean $\mathbf{m} = m(x_i)$ and the noise matrix $\mathbf{V}$ without loss of generality. Training a GP means sampling from or maximizing $\log(p(\mathbf{y}|\phi))$ with respect to the hyperparameters $\phi$. The problem of interest arises from the fact that $\mathbf{K} \in \mathbb{R}^{|\mathcal{D}| \times |\mathcal{D}|}$, where $|\mathcal{D}|$ is the size (cardinality) of the dataset. The prior covariance matrix $\mathbf{K}$ has to be stored and inverted (or, equivalently, a linear system solved). This leads to $\mathcal{O}(|\mathcal{D}|^2)$ storage complexity and $\mathcal{O}(|\mathcal{D}|^3)$ time complexity. In addition, calculating $\log(|\mathbf{K}|)$ also scales approximately with complexity $\mathcal{O}(|\mathcal{D}|^3)$.

Once hyperparameters are found, the posterior probability $f(x^*) = f^*$ can be calculated as

$$p(f^*|\mathbf{y}) = \mathcal{N}\Big(m(x^*) + k(\mathbf{X}, x^*)^T \mathbf{K}^{-1}(\mathbf{y} - \mathbf{m}),$$
$$k(x^*, x^*) - k(\mathbf{X}, x^*)^T \mathbf{K}^{-1} k(\mathbf{X}, x^*)\Big), \tag{2}$$

where $\mathbf{X}$ is the matrix containing all $x_i \ \forall i \in \{1, 2, ..., |\mathcal{D}|\}$. If $\mathbf{K}^{-1}$ was stored during training, this is a quick operation, but this is rarely the case because the inversion is generally avoided due to accrued inaccuracies. Otherwise, this operation will also require solving a linear system.

Both the unfavorable storage and time complexity of training and prediction traditionally limit the application of GPs to small and medium-sized datasets. Although approximation methods exist and are often applied, they typically affect the GP's accuracy and, worse still, limit its customization flexibility — arbitrary non-stationary kernels and heteroscedastic parametric noise models.

## 4. Method

To recap, the core idea motivating the proposed method *gp2Scale* is that GPs may well scale to large datasets if the kernel design is customized to allow the discovery of a sparse covariance matrix. Therefore, the kernel has to possess the ability to return zero when two function values are deemed independent. This can be achieved through compact support of the kernel functions. In the stationary case, this is commonly referred to as covariance tapering in the literature (Zhang & Du, 2008; Kaufman et al., 2008; Furrer et al., 2006) and has been widely criticized for excluding *far-field interactions* — those not determined solely by proximity under a distance metric. In stationary datasets lacking far-field dependencies, covariance tapering is a clever and efficient method to scale GPs while maintaining exactness. Recently, non-stationary kernels have gained popularity, offering flexible ways to encode distance-unrelated (far-field) dependencies. Following this logic, if we can equip a GP with a flexible, non-stationary, compactly supported kernel, we may recover a sparse covariance matrix while accurately modeling near and far-field interactions, thereby enabling accurate predictions and uncertainty quantification. In the

following, we introduce a set of kernels that possess the required properties: flexibility, non-stationarity, and compact support.

## 4.1. Wendland-Style Kernels via the Product of Kernels

Wendland kernels (Wendland, 1995) are a particularly prominent class of stationary, compactly supported kernels. In our implementation and in the experiments, we will heavily rely on the particular Wendland kernel

$$
k_{\mathcal{W}}(x_i, x_j; r_0) = \begin{cases} \left(1 - \frac{\|x_i - x_j\|}{r_0}\right)^8 \times \left(32 \left(\frac{\|x_i - x_j\|}{r_0}\right)^3 \right. \\ \left. + 25 \left(\frac{\|x_i - x_j\|}{r_0}\right)^2 + \frac{8\|x_i - x_j\|}{r_0} + 1\right) \\ \quad \text{if } \|x_i - x_j\| < r_0, \\ 0, \text{otherwise.} \end{cases}
$$
(3)

and small variations thereof, where $r_0$ is the radius of support, and $||\cdot||$ is the Euclidean norm. Since any product of symmetric positive semi-definite functions — the core property all kernels have to be endowed with — is symmetric and positive semi-definite, we can freely formulate kernels of the type

$$
k(x_i, x_j) = k_{nonstat}(x_i, x_j) k_{\mathcal{W}}(x_i, x_j) \tag{4}
$$

with arbitrary $k_{nonstat}$ to define non-stationary compactly supported kernels. This particular kernel, however, will mute far-field interactions outside of $k_{\mathcal{W}}$'s support.

## 4.2. A Non-Stationary Wendland Kernel via Convolution

To derive a non-stationary extension of the stationary Wendland kernel (3), we take advantage of the fact that the convolution of two kernels

$$
k_c(x_i, x_j) = \int_{\mathbb{R}^n} k(x_i, x) k(x_j, x) dx \tag{5}
$$

results in a valid kernel (Paciorek & Schervish, 2006; Risser & Turek, 2020; Higdon et al., 1999). We can use this fact to propose the kernel

$$
k(x_i, x_j) =
$$
$$
\sigma_s(x_i)\sigma_s(x_j)\frac{|\mathbf{\Sigma}(x_i)|^{1/4}\,|\mathbf{\Sigma}(x_j)|^{1/4}}{\left|\frac{\mathbf{\Sigma}(x_i)+\mathbf{\Sigma}(x_j)}{2}\right|^{1/2}} k_{\mathcal{W}}\left(\sqrt{Q(x_i, x_j)}\right),
$$
(6)

where $\sigma_s$ is the non-constant signal standard deviation, $\mathbf{\Sigma}(x)$ is the anisotropic non-constant length scale as a function on the input set, here $|\cdot|$ denotes the determinant, $k_{\mathcal{W}}$ is a Wendland kernel (for instance the one defined in 3),

and $Q(x_i, x_j) = (x_i - x_j)^\top \left(\frac{\mathbf{\Sigma}(x_i)+\mathbf{\Sigma}(x_j)}{2}\right)^{-1} (x_i - x_j)$. Although $Q(x_i, x_j)$ is not a valid distance metric (it violates the triangle inequality), this kernel is positive semi-definite, as demonstrated by (Paciorek & Schervish, 2006). This construction yields a highly flexible kernel, although far-field interactions are still neglected outside of the support of $k_{\mathcal{W}}$.

## 4.3. The Bump-Function Kernel

To include far-field interactions, we take advantage of so-called bump functions

$$
b(x, x_i) = \begin{cases} a \exp\left\{\beta\left[1 - (1 - \frac{|x-x_i|^2}{r^2})^{-1}\right]\right\} & \text{if } |x - x_i|^2 < r \\ 0 & \text{else,} \end{cases}
$$
(7)

where $a$ is the amplitude, $\beta$ is an optional shape parameter, and $r$ is the bump function radius. The bump function is compactly supported but not a valid kernel. However, $k(x_i, x_j) = g(x_i)g(x_j)$, is indeed a valid kernel for any function $g$ (including bump functions, see Theorem 1 in Appendix A) and a convenient way to create non-stationary kernels with flexible signal variances when combined by-product with a stationary kernel (Noack & Sethian, 2022). If we now consider $g$ to be the bump function in Equation (7), we recover a non-stationary and compactly supported kernel. This kernel lacks flexibility because $g(x_i)g(x_j)$ yields a rank 1 Gram matrix (for any $g$) and can therefore only turn data points "on" or "off"; for $g = b$ this also can inadvertently turn off covariances along the diagonal when pairs of points are located outside of the support of any bump functions, leading to nonphysical behavior in which a data point is not correlated with itself. Both issues can be avoided by considering the sum

$$
k(x_i, x_j) = g(x_i)g(x_j) + \sigma_s^2 k_{\mathcal{W}}(x_i, x_j), \tag{8}
$$

where local interactions are now preserved, and the associated Gram matrix has full rank. To flexibly model far-field interactions, we define $g(x) = \sum_p^P b(x, x_p)$ which means $g(x_i)g(x_j) = \sum_{pq}^P b(x_i, x_p)\, b(x_j, x_q)$. Far-field interactions are still rank 1 and can only enable or disable covariances for data points with respect to all other points (see Appendix C.1). Higher-order interactions, however, can be modeled by considering $\sum_u^U g_u(x_i)g_u(x_j)$, which has rank $\leq U$. The implementation of those adaptations results in the kernel

$$
k(x_i, x_j) = k_{core}(x_i, x_j)\left(\sum_u^U g_u(x_i)g_u(x_j) + k_{\mathcal{W}}(x_i, x_j)\right),
$$
(9)

where we included the optional product with an arbitrary user-defined domain-customized core kernel $k_{core}$. This is

the first kernel in the class that can flexibly model far-field interactions, since the bump functions can be located anywhere in the input space. It also allows for higher-order interactions for $U > 1$: a point set A might be correlated with a point set B but not with a point set C, while B and C are highly correlated. We included a graphical illustration of the covariance matrix for this kernel in Appendix C.1. The shape, amplitudes, and radii of the bump functions can be held constant or be defined parametrically over the input domain, allowing control over the number of hyperparameters. The positions of the bump functions can be trained or fixed to a grid or to a subset of data point locations. An alternative option is to use clustering and position bumps at cluster centers.

## 4.4. Collapsing Bumps into Deltas

The introduced bump-function-style non-stationary kernels enable far-field interactions as defined via the collective support of the bumps. While this is intuitive, it may lack flexibility and alter the covariance structure imposed by $k_{core}$ because of the bump function's smooth shape. In those cases, we may collapse the radii in Equation (7) to zero, which effectively results in deltas $\delta(x, x_i) = 1$ if $x = x_i$ and 0 otherwise. This allows us to consider distance-unrelated non-stationary interactions very flexibly. More specifically, the kernel

$$k_d(x_i, x_j) = \sum_p^{|\mathcal{D}|} g_p(x_i)g_p(x_j), \tag{10}$$

where $g_p(x) = \sum_q^Q \delta(x, x_q)$ defines a non-stationary and compactly supported kernel in which, in principle, each data point can choose to have non-zero covariances with an arbitrary set of other points. For numerical stability, this kernel can be added to a Matérn kernel — to make the covariance matrix diagonally dominant if needed by downstream solvers — and multiplied by any user-specified domain-motivated core kernel. In this case, we obtain, similar to (9), $k(x_i, x_j) = k_{core}(k_{\mathcal{W}} + \sum_p^{|\mathcal{D}|} g_p(x_i)g_p(x_j))$. The large number of terms in the two sums might worry some readers about the required hyperparameters for this kernel, but rules for how to choose the positions of the deltas can often be encoded parametrically with very few hyperparameters. For example, one can mimic a flexible nearest-neighbor approach by introducing one additional hyperparameter: the radius of the neighbors, or the number of neighbors (which may vary as a function of the input space). Overall, using deltas instead of bumps allows for a mask that leaves the core kernel unchanged within the support, while effectively maximizing sparsity. On the flip side, differentiability is lost.

## 4.5. An Extension for Small Length Scales in $k_{core}$

The kernel in Equation (9) enables us to use bump functions as a type of mask that activates covariances specified by the non-stationary, locally interacting core kernel $k_{core}$. If the length scales of that core kernel are comparably small, far-field interactions will be muted. Separating local and far-field interactions via $k(x_i, x_j) = k_{core_1}k_{\mathcal{W}} + k_{core_2}\sum_p g_p(x_i)g_p(x_j)$ will allow far-field interaction to remain active, even for small $k_{core_1}$ length scales. $k_{core_1}$ and $k_{core_2}$, in this case, may only differ by their respective hyperparameters. An example kernel of this kind can be found in Appendix C.2. This kernel circumvents the problem of muted far-field covariances by tying the bump-function kernel to a globally supported $k_{core_2}$, which is, however, only active within the support of the bump functions. This allows the influence of the bump functions to have a truly non-local component.

## 4.6. Distributed Computing and Block MCMC

With the class of flexible non-stationary and compactly supported kernels in place, the remaining building blocks of the *gp2Scale* framework are the distributed-computing framework and the block-MCMC. Although both building blocks are crucial, they are much less involved compared to the kernel designs, which are the core methodological advancement. Since we are calculating the covariance matrix of an exact Gaussian process (GP), we need to distribute that calculation across as many nodes (ideally GPUs) as possible. The dataset is divided into similarly-sized blocks, which are then sent to the distributed workers for processing. There, the square covariance block-matrices for data-block pairs are computed and returned in sparse COO format, thereby reducing communication load. The matrix is assembled and cast to CSR format on the host node, where the solutions to $\mathbf{K}^{-1}\mathbf{y}$ and $\log(|\mathbf{K}|)$ are subsequently computed. See Appendix B.7 for details. Assuming that the kernel identified a sufficiently sparse structure, both operations are very fast (see Appendix B.6). This makes the $\mathcal{O}(|\mathcal{D}|^2)$ scaling of the computation of the covariance matrix the most costly part. Note that one might mistakenly assume that only non-zero elements of the covariance matrix need to be computed, which would lead to better scaling. However, non-zero elements are not predetermined but a result of the kernel evaluation, so all covariance matrix entries must be computed. Assuming sufficient resources are available, the trivially parallelizable covariance matrix computation results in nearly constant wall-clock time. Given the nature of the hyperparameters and their independence, they can be sampled in separate blocks during MCMC, potentially leading to faster convergence. However, evaluating each MCMC block requires an additional solve and a log-likelihood evaluation.

## 5. Experiments and Results

In what follows, we compare the approximation performance of *gp2Scale* to state-of-the-art community-accepted implementations of SVGP, VNNGP, SKI, and the Vecchia approximation. We acknowledge that extensions of some of these methods exist and might lead to slightly better performance. However, these same advances often create additional degrees of freedom, requiring us to make choices that affect the validity and generalizability of our tests. Since we are comparing an exact GP with approximations, the needed computing resources are vastly different, and comparing computing times and needed architecture is therefore meaningless. However, we report computational details in the Appendix for the benefit of the reader and future users. *gp2Scale* is implemented and available to users as part of the open-source *gpCAM* Python package (`https://gpcam.lbl.gov/`). We report the performance scores as means and standard deviations across 10 independent runs for all computational experiments, except for the final production run on 10 million data points, which was limited by computational constraints. We present the *RMSE* and the *CRPS* for the evaluation of prediction and uncertainty quantification abilities of the proposed algorithm. The results of the best-performing code are highlighted in bold. All run scripts, training, and test data can be found in the shared repository (see Reproducibility statement).

### 5.1. A 1-Dimensional Synthetic Function

We want to start our computational experiments with the 1-dimensional synthetic function

$$f_1(x) = \sin(5x) + \cos(20x) + 2(x - 0.4)^2 \cos(400x)$$

for easy visual inspection of the solutions. In *gp2Scale*, we used the kernel in Equation 6. At 2000 training data points, this example can be computed with a standard base-GP with Matèrn $\nu = 3/2$ for comparison. The result is shown in Figure 1. We also implemented the kernel for *gp2Scale* in SVGP, but the default Adam optimizer failed to find a high-quality solution. The most striking takeaway from this simple test is that all approximate methods smooth out local characteristics of the complex and non-stationary test function (see Table 1). The implementation details for the competing methods can be found in the Appendix and the shared repository.

### 5.2. Topography

In this example, we train a GP regressor on 20,000 data points representing the United States' topography. The test set comprises 5000 randomly chosen data points. While still low-dimensional, this dataset is challenging due to its high degree of non-stationarity. The dense sampling leads to the identification of substantial sparsity in the covari-ance matrix. The non-stationarity in the data requires the use of a customized kernel. Since *gp2Scale* is agnostic to the core user-defined kernel design, we observe superior performance using kernel (6) and (9) (see Table 2). The implementation details for the competing methods can be found in the Appendix and the shared repository.

### 5.3. 8-Dimensional California Housing Dataset

This dataset comprises 20,000 data points in eight dimensions, along with their corresponding labels, which are housing prices from California (`https://www.dcc.fc.up.pt/~ltorgo/Regression/cal_housing.html`). The test data set contains 640 points. The complexity of this dataset lies in the fact that the high-dimensional space induces data scarcity, making the discovery of naturally occurring sparsity in the covariance structure more difficult. *gp2Scale* was executed using kernel (3) with axially-anisotropic distances (Automated Relevance Determination, ARD (Williams & Rasmussen, 1995)) and performed competitively (see Table 3) nonetheless. SKI used an additive kernel because the dataset's dimensionality exceeds the recommended value of 4 (Wilson & Nickisch, 2015). The implementation details for the competing methods are available in the Appendix and the shared repository.

### 5.4. 60,000 MNIST Handwritten Digits

For this test, we extracted 70,000 handwritten digits from the MNIST dataset (http://yann.lecun.com/exdb/mnist/), provided as 28 by 28 pixel arrays, and randomly divided the set into 60,000 training samples and 10,000 test samples. For simplicity, we turn this classification problem into a regression problem of the function $f(x) = \Pr(y = 5)$. The labels are then 1 if the digit value is 5 and 0 otherwise. Although this is somewhat of a departure from common practice, treating this as a regression problem of a probability despite it being a multi-class classification, highlights the scalability and agnosticism to the input set of the proposed methodology. As a performance measure, we are using the well-established Brier score. For this test, we skipped Vecchia because it would have taken a substantial revamp of the existing package to work with this dataset. The Vecchia R package (https://github.com/katzfuss-group/GPvecchia) is tailored for spatial statistics. The combination of variational inference, inducing points, and a notion of neighboring points led to poor performance for VNNGP. SVGP performed competitively. SKI cannot be applied to this example due to the size of the $28 \times 28$-dimensional local grid. *gp2Scale* is an exact GP, which means there are no restrictions on the type of input space. For the *gp2Scale* run, we used the kernel $k_{\mathcal{W}} k_d$ — the product of kernels (3) and (10). For all runs, we used the $l_1$ norm as the distance metric. Distances are well-known to collapse to a narrow

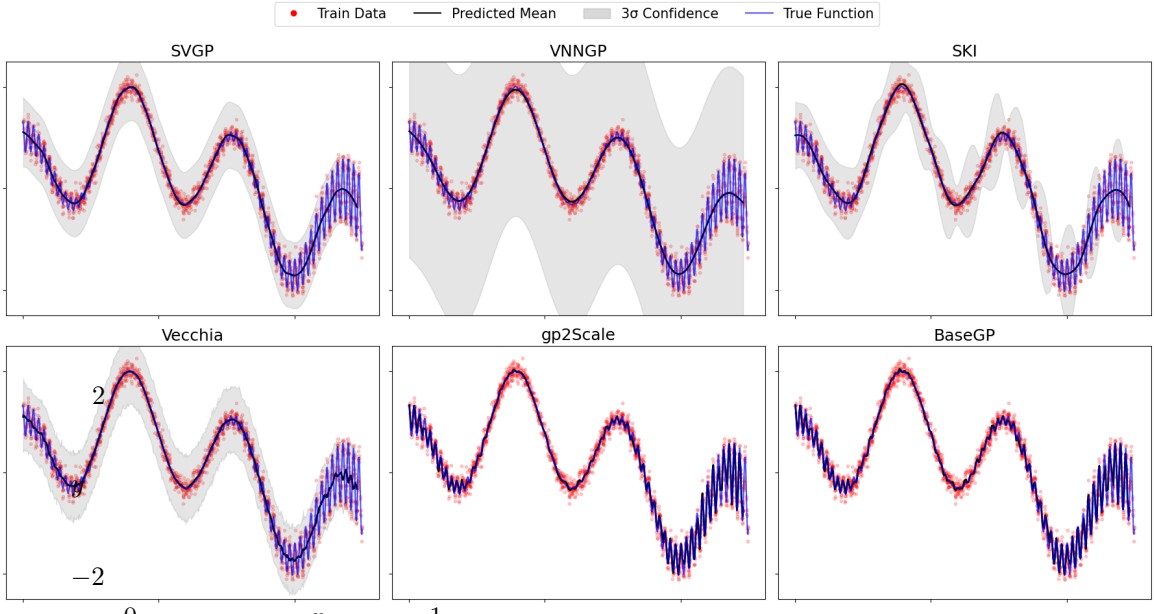

*Figure 1.* Approximation performance of VNNGP, SVGP, SKI, Vecchia, *gp2Scale*, and a base regular GP for comparison, which is, in most examples we consider, computationally prohibitive. The ground truth is depicted in blue; red dots represent the training data, and the posterior mean is shown in black. VNNGP, SVGP, and SKI oversmooth and cannot adequately recover local oscillations. The Vecchia approximation shows some local oscillation, although degraded. *gp2Scale* best preserves local variations and sharp transitions, and offers reliable uncertainty quantification (UQ) very similar to the base regular GP. Note that for all figures the posterior standard deviation of $p(\mathbf{f})$ is displayed, not of $p(\mathbf{y})$, leading to correct, vanishingly small uncertainties for the regular GP and *gp2Scale*, which is expected given the dense data distribution. The competing methods are clearly overestimating the uncertainty. The noise, in this example, is homoscedastic and constant across the tested methodologies.

*Table 1.* 1-Dim. Synthetic Experiment.

| Name | SVGP | VNNGP | SKI | Vecchia | Base GP | *gp2Scale* w/(6) |
|------|------|-------|-----|---------|---------|------------------|
| RMSE | 0.19±5.4e-4 | 0.20±8.9e-5 | 0.19±1.3e-4 | 0.20±0.025 | 0.109±4.4e-4 | **0.107±9.0e-5** |
| CRPS | 0.11±3.7e-4 | 0.21±1.6e-5 | 0.11±2.4e-4 | 0.11±0.013 | 0.07±11.7e-5 | **0.06±14.9e-5** |

range in high-dimensional spaces; it is therefore important to plot the distributions of the pairwise distances and set sampling ranges appropriately. See Table 4 for the results.

### 5.5. 3-Dimensional Temperature Dataset with 10 Million Points

The last example is specifically designed to illustrate the scaling potential of *gp2Scale*. The training dataset comprises 10 million measured temperatures across the United States, spanning approximately 10 years (Menne et al., 2012). We ran this experiment on 1024 A100 GPUs on NERSC's Perlmutter supercomputing system. Within the available computing time, we managed to run circa 500 MCMC iterations on a 1-million-data-point representative subset of the data and 100 MCMC iterations on the full dataset, leading to a well-performing but not yet optimal model; however, we obtained competitive results, beating the best competitor (Vecchia) by a slight margin nonetheless. This test demonstrates that truly massive exact GPs are

possible. One MCMC iteration took 477 seconds to complete. From this, we can deduce that a full run from scratch might take about a week of runtime. Although this might sound like a long time, it is in line with the training times of some large neural networks or LLMs. When a highly customizable exact GP is needed for a large data set, *gp2Scale* can deliver superior performance when computing time and resources are available. The results are summarized in Table 5. For this dataset, we are only reporting the RMSE of one execution due to computing resource limitations.

## 6. Discussion and Conclusion

In this manuscript, we propose a new methodology, termed *gp2Scale*, for scaling exact Gaussian process regression up to (and possibly beyond) 10 million data points. We have shown that the method performs competitively with state-of-the-art approximation methods. At the core of the methodology lies the assumption that GPs are not naturally

*Table 2.* Topography.

| Name | SVGP | VNNGP | SKI | Vecchia | *gp2Scale* w/(6) | *gp2Scale* w/(9) |
|------|------|-------|-----|---------|------------------|------------------|
| RMSE | $266.5 \pm 2.1$ | $236.0 \pm 0.1$ | $206.3 \pm 0.1$ | $150.0 \pm 0.2$ | $136.3 \pm 0.17$ | **$136.1 \pm 1.03$** |
| CRPS | $148.0 \pm 3.0$ | $175.5 \pm 0.1$ | $117.3 \pm 0.3$ | $78.4 \pm 0.10$ | **$63.8 \pm 0.81$** | $74.6 \pm 2.96$ |

*Table 3.* 8-Dim. CA Housing.

| Name | SVGP | VNNGP | SKI | Vecchia | *gp2Scale* w/(3) |
|------|------|-------|-----|---------|------------------|
| RMSE | $0.60 \pm 3.0\text{e-}4$ | $0.66 \pm 4.0\text{e-}4$ | $0.71 \pm 1.9\text{e-}4$ | $0.60 \pm 0.15$ | **$0.49 \pm 2.6\text{e-}3$** |
| CRPS | $0.35 \pm 1.1\text{e-}4$ | $0.41 \pm 1.9\text{e-}4$ | $0.50 \pm 2.6\text{e-}4$ | $0.31 \pm 0.10$ | **$0.27 \pm 9.5\text{e-}3$** |

*Table 4.* MNIST Dataset.

| Name | SVGP | VNNGP | SKI | Vecchia | *gp2Scale* w/$k_{\mathcal{W}} k_d$ |
|------|------|-------|-----|---------|------------------------------------|
| BRIER | $0.033 \pm 2.8\text{e-}5$ | $0.052 \pm 2.9\text{e-}3$ | NA | NA | **$0.018 \pm 0.002$** |

*Table 5.* 3-Dim. Temperatures.

| Name | SVGP | VNNGP | SKI | Vecchia | *gp2Scale* w/(3) |
|------|------|-------|-----|---------|------------------|
| RMSE | 5.90 | 5.21 | NA | 2.8602 | **2.8509** |

dense; rather, standard kernels impose density on the covariance matrix, leading to the well-known scaling challenges. Flexible, non-stationary, and compactly supported kernels, in contrast, allow the GP to discover sparsity naturally. This stands in contrast to competing approximate methods, in which sparsity in the covariance or precision matrix is induced by user-specified choices, such as the number of inducing points or neighbors. The main advantage of our method is that the GP remains exact, allowing for superior prediction performance in many cases, but more importantly, imposes absolutely no restrictions on user-required GP customizations. The proposed kernels can all be viewed as masks, enabling sparsity to be discovered, given a user-defined "core" kernel.

However, we see *gp2Scale* not as a blanket solution but as a part of a practitioner's arsenal when tackling large-scale GPs. We acknowledge that approximate methods perform remarkably well in certain situations at low computational cost. The Vecchia approximation, for instance, was hard to beat on the California Housing dataset. The dataset is relatively high-dimensional, leading to sparsely distributed data points and making it challenging to discover naturally occurring sparsity in the covariance structure. In addition to the data scarcity, non-stationarity plays only a minor role, limiting the value of our methodology. VNNGP can be seen as a special case of Vecchia and shows mixed performance. SVGP often oversmooths, resulting in subpar performance. SKI had similar issues in our tests.

As a summary of our tests, we recommend using approximate methods when time and hardware availability are limited. For simple functions, inducing point methods are hard to beat, while in a spatial context without significant non-stationarity, Vecchia approximations stood out. *gp2Scale* is best used for sophisticated, highly customizable GPs on densely sampled, non-stationary functions.

## Author Contributions

M.M.N.: Ideation, Kernel derivation, Data curation, Performance comparisons, gp2Scale software development, Manuscript; M.D.R.: Ideation, Kernel derivation, Data curation, Performance comparisons, gp2Scale software development, Manuscript; H.L.: Ideation, Kernel derivation; V.T.: Data curation, Performance comparisons, Manuscript; R.P.: gp2Scale software development

## Acknowledgments

This work was supported by

- The Center for Advanced Mathematics for Energy Research Applications (CAMERA), which is jointly funded by the Advanced Scientific Computing Research (ASCR) and Basic Energy Sciences (BES) within the Department of Energy's Office of Science, under Contract No. DE-AC02-05CH11231.

- The U.S. Department of Energy, Office of Science, Office of Advanced Scientific Computing Research's Applied Mathematics Competitive Portfolios program under Contract No. AC02-05CH11231.

- This research was supported in part by the U.S. Department of Energy, Office of Science, Office of Advanced

Scientific Computing Research's Applied Mathematics program under Contract No. DE-AC02-05CH11231 at Lawrence Berkeley National Laboratory; and U.S. National Science Foundation NSF-DMS 2412403 at Rice University.

- The Director, Office of Science, Office of Biological and Environmental Research of the U.S. Department of Energy under the Regional and Global Model Analysis program and the CASCADE Scientific Focus Area (Contract No. DE-AC02-05CH11231).

This research used resources of the National Energy Research Scientific Computing Center (NERSC), a U.S. Department of Energy Office of Science User Facility located at Lawrence Berkeley National Laboratory, operated under Contract No. DE-AC02-05CH11231, using NERSC awards ERCAP0031656, ERCAP0032615 and ERCAP0032229.

## Reproducibility statement

To ensure full reproducibility, we created a GitHub repository containing all the code and instructions to reproduce the results `https://github.com/MarcusMNoack/gp2Scale`. All data are publicly available, and a link is included in the repository.

## Conflict of Interest Disclosure

The authors declare no conflicts of interest or ethics violations.

## Impact Statement

This paper presents work whose primary goal is to advance scalable Gaussian process methodology for scientific machine learning. The societal consequences are largely those associated with progress in probabilistic modeling more broadly; however, a few points merit specific attention. The most direct positive impact of gp2Scale is its preservation of rigorous, calibrated uncertainty quantification (UQ) at scales previously accessible only to approximate methods. In scientific domains — climate modeling, geospatial analysis, materials discovery, autonomous experimentation — decisions informed by GP posteriors are only as trustworthy as the UQ those posteriors provide. By enabling exact GPs at the scale of millions of observations, this work reduces the risk of overconfident predictions that can arise from variational or inducing-point approximations, with potential downstream benefits for data-driven scientific decision-making. A potential concern is computational cost. Scaling to 10 million data points required 1024 A100 GPUs on a national supercomputing facility, and a full production run is estimated at approximately one week of wall-clock time. This energy and resource footprint should not be dismissed. We note, however, that gp2Scale is not intended to replace efficient approximate methods for routine tasks; rather, it is positioned as a high-fidelity tool for situations where exactness and customizability are scientifically necessary — analogous to the role of high-fidelity simulation in computational science. Users should weigh this trade-off deliberately. The software is released as part of the open-source gpCAM package, making these capabilities broadly accessible beyond institutions with dedicated HPC resources, albeit with the practical constraint that large-scale runs will require commensurate hardware.

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

## A. Used Theorems

**Theorem 1.** *Let $k(x_1, x_2)$ be a valid kernel, then $f(x_1)f(x_2)k(x_1, x_2)$ is also a valid kernel. Here, $f(x)$ is an arbitrary function over the input set.*

*Proof.* Since $k$ is a valid kernel,
$$\sum_i^N \sum_j^N c_i\, c_j\, k(x_i, x_j) \geq 0\, \forall N,\, x \in \mathbb{R}^N,\, \mathbf{c} \in \mathbb{R}^N$$
$$\Rightarrow \sum_i^N \sum_j^N f_i\, f_j\, c_i\, c_j\, k(x_i, x_j)$$
$$\geq 0\, \forall N,\, x \in \mathbb{R}^N,\, \mathbf{c} \in \mathbb{R}^N\, \mathbf{f} \in \mathbb{R}^N$$
$$\Rightarrow \sum_i^N \sum_j^N c_i\, c_j\, f(x_i)\, f(x_j)\, k(x_i, x_j) \geq 0\, \forall N,\, x \in \mathbb{R}^N$$
$\square$

## B. Codes, Computing Architecture and Compute Times

We used PyTorch implementations for SVGP(`https://docs.gpytorch.ai/en/latest/variational.html`) and SKI(`https://docs.gpytorch.ai/en/v1.6.0/examples/02_Scalable_Exact_GPs/KISSGP_Regression.html`), and the VN-NGP implementation following the example of GPyTorch (`https://docs.gpytorch.ai/en/v1.13/examples/04_Variational_and_Approximate_GPs/VNNGP.html`). The Vecchia R package we used for the first two tests can be found here `https://cran.r-project.org/web/packages/GpGp/index.html`. All other Vecchia tests were run with code found here `https://github.com/katzfuss-group/scaledVecchia/tree/master`. *gp2Scale* is implemented as part of the *gpCAM*, `https://gpcam.lbl.gov/` open-source Python package. SVGP, SKI, and VNNGP were all trained with the Adam optimizer. Vecchia is trained with MLE-II; gp2Scale is always trained via MCMC for assessment and propagation of hyperparameter uncertainties. While this might lead to some discrepancies in method comparisons, these mechanisms are baked into the software packages, and the observed performance differences are unlikely to be caused by them. If this were the case, it should be seen as a strength of methods compatible with MCMC.

### B.1. 1-Dim. Synthetic

The SVGP, SKI, and VNNGP for the one-dimensional test were run on the T4 GPU on Google Colab, which is equipped with an Intel Xeon CPU with two vCPUs (virtual CPUs) and 13GB of RAM, and one T4 GPU. SVGP was run with 10 inducing points, VNNGP used 2000 inducing points (full training dataset) and 50 neighbors, and SKI used 20 local grid points. The Vecchia approximation code was run on a single core of a local 24-core node with 128 GB of shared RAM. Total run time was 14 seconds (11.4 sec-

onds for training and 2.6 seconds for predictions at the test points). We used the default settings of 30 conditioning points per data point. *gp2Scale* was run on a single-node Intel Core i9-9900KF CPU. The computation time for all methods was on the order of minutes.

### B.2. Topography

The SVGP, SKI, and VNNGP for the topography test were run on the T4 GPU on Google Colab, which is equipped with an Intel Xeon CPU with two vCPUs (virtual CPUs) and 13GB of RAM, and one T4 GPU. SVGP was run with 100 inducing points, VNNGP used 20,000 inducing points (full training dataset) and 50 neighbors, and SKI used 30 local grid points per dimension. Vecchia ran on a single core of a local 24-core node with 128 GB of shared RAM. Total run time was 494 seconds (442 seconds for training and 52 seconds for predictions at the test points). We used the default settings of 30 conditioning points per data point. *gp2Scale* was run on 15 A100 GPUs and ran in about 1 hour. The approximate methods ran in about 15 minutes each.

### B.3. 8-Dim. CA Housing

The SVGP, SKI, and VNNGP for the housing test were run on the T4 GPU on Google Colab, which is equipped with an Intel Xeon CPU with two vCPUs (virtual CPUs) and 13GB of RAM, and one T4 GPU. The approximate methods ran in 30 minutes to about an hour. SVGP was run with 100 inducing points; VNNGP used 20,000 inducing points (full training dataset) and 50 neighbors; and SKI used 30 local grid points per dimension (an additive kernel due to the dimensionality). Vecchia ran on a single core of a local 24-core node with 128 GB of shared RAM. Total run time was 349 seconds (334 seconds for training and 15 seconds for predictions at the test points). We used the default settings of 30 conditioning points per data point. *gp2Scale* was run on 15 A100 GPUs and ran in about 1-4 hours (based on the number of MCMC iterations).

### B.4. MNIST Dataset

The SVGP, SKI, and VNNGP for the MNIST test were run on a dedicated NERSC Perlmutter node, which is equipped with an Intel 2x AMD EPYC 7763 CPU. The approximation codes did not utilize the GPU. The approximate methods ran in about 2 hours. SVGP used 500 inducing points. A larger number decreased prediction accuracy. VNNGP used 20,000 inducing points (a third of the dataset) and 1,000 neighbors, where decreasing the number of inducing points resulted in the method reverting to mean prediction. *gp2Scale* was run on 15 A100 GPUs and ran in about 1-4 hours (based on the number of MCMC iterations).

### B.5. 3-Dim. Temperatures.

The SVGP, SKI, and VNNGP for the 3-Dim.-Temperatures test were run on a dedicated NERSC Perlmutter node, which is equipped with an Intel 2x AMD EPYC 7763 CPU. The approximation codes did not utilize the GPU. SVGP ran in about 4 hours. SVGP was run with 300 inducing points (larger numbers exceeded RAM threshold). SKI ran out of memory even with only 4 grid points per dimension. VNNGP used 10,000 inducing points with 1,000 neighbors, and took approximately 2 hours. Vecchia ran on a single core of a local 104-core node with 1 TB of shared RAM. Total run time was 5.2 hours (most for training and less than one minute for predictions at the test points). We used 10 conditioning points per data point to minimize computational time. *gp2Scale* was run on 1024 A100 GPUs, which led to an execution time of about 477 seconds per MCMC iteration. We expect a full run to use about 1000 MCMC iterations. This is because the hyperparameters have physical meaning and can be initialized quite close to their final values.

### B.6. Compute Times Across Different Problem Sizes

Computing times of one MCMC iteration (covariance calculation, MINRES solve, and log-det calculation) for the kernel in Eq. (9) and a generic synthetic dataset on 16 A100 GPUs. This demonstrates that, if sufficient sparsity is discovered, the calculation of the covariance matrix is the most expensive operation of the compute pipeline, especially as problem size increases.

| Dataset Size | Sparsity | Covariance | MINRES | LOGDET | Total |
|---|---|---|---|---|---|
| 50000 | 2.01E-05 | 1.11905 | 0.00666 | 0.92695 | 2.05560 |
| 50000 | 4.02E-05 | 1.10673 | 0.05059 | 1.09341 | 2.25396 |
| 50000 | 2.66E-04 | 1.13132 | 0.08509 | 0.96690 | 2.19455 |
| 100000 | 1.21E-05 | 2.03161 | 0.06763 | 0.97226 | 3.07747 |
| 100000 | 7.15E-05 | 2.03520 | 0.03350 | 0.92840 | 3.01514 |
| 100000 | 9.50E-05 | 2.02798 | 0.10975 | 0.98395 | 3.14211 |
| 200000 | 6.92E-06 | 10.1834 | 0.01366 | 0.92315 | 11.1279 |
| 200000 | 9.75E-05 | 10.2132 | 0.01185 | 0.92963 | 11.1715 |
| 200000 | 6.40E-04 | 10.9462 | 6.97306 | 1.06100 | 18.9802 |

### B.7. Distributed Computing Pipeline

The distributed-computing pipeline is shown in Figure 2. We provide a detailed, practical explanation here. The computational process starts by dividing the datasets into a number of subsets of a given size that depends on the distributed compute architecture. The distributed computing is handled via Python's DASK. The data subset size per worker depends on the compute resources each one has available and the particular kernel definition. The worker will use the kernel to calculate a block of the covariance matrix. Maximum RAM usage should not exceed the available RAM per

worker and should optimally utilize the available compute. For example, the A100 GPUs we run many of our models on are optimally utilized at a block size of 15000 points. This number can often be smaller for CPU workers. Once the dataset is distributed to the workers, each worker calls the kernel function and computes the corresponding block of the covariance matrix. Still on the worker, the matrix block is cast into the sparse COO format to minimize communication burden between the worker and host nodes. Once all blocks are computed, the full sparse covariance matrix is assembled on the host node. Sparse linear algebra follows and may use the same workers. A user should ensure that there are at least as many blocks to compute as there are GPUs available, for optimal hardware utilization. When more blocks have to be computed than GPUs are available — this is the most common case — work will be optimally scheduled automatically. The object $K^{-1}\mathbf{y}$, where $K$ is the covariance matrix, is stored for fast posterior mean computation. The described process is repeated for each MCMC iteration for the training.

## C. More Information on Kernels

### C.1. Covariance Matrix Illustration

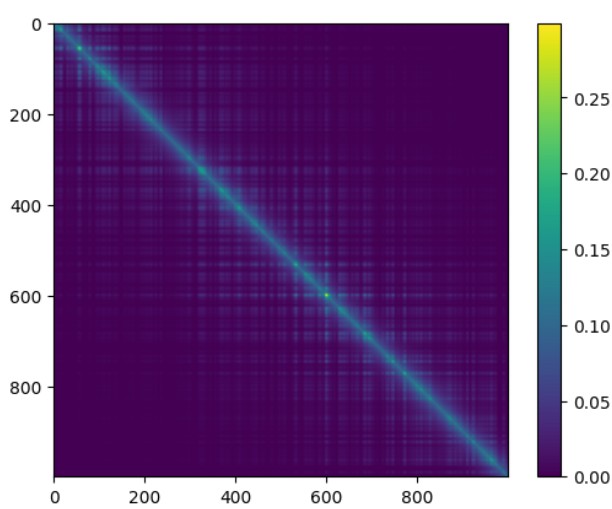

*Figure 3.* Graphical illustration of the covariance matrix for a one-dimensional problem using kernel (9) for $U = 1$ and $k_{core} = 1$, In particular, the far-field term $g(x_i)g(x_j)$ has rank 1 and therefore the associated interactions can only be turned "on" or "off" which is highlighted by a box pattern in the covariance matrix

### C.2. Combination Kernel

Here, we create a combination kernel from the core kernel

$$k_{core}(x_i, x_j) = \sigma(x_i)\sigma(x_j)\frac{|\mathbf{\Sigma}(x_i)|^{1/4}\,|\mathbf{\Sigma}(x_j)|^{1/4}}{\left|\frac{\mathbf{\Sigma}(x_i)+\mathbf{\Sigma}(x_j)}{2}\right|^{1/2}}$$
$$k_{\mathcal{M}}\left(\sqrt{Q(x_i, x_j)}\right), \tag{11}$$

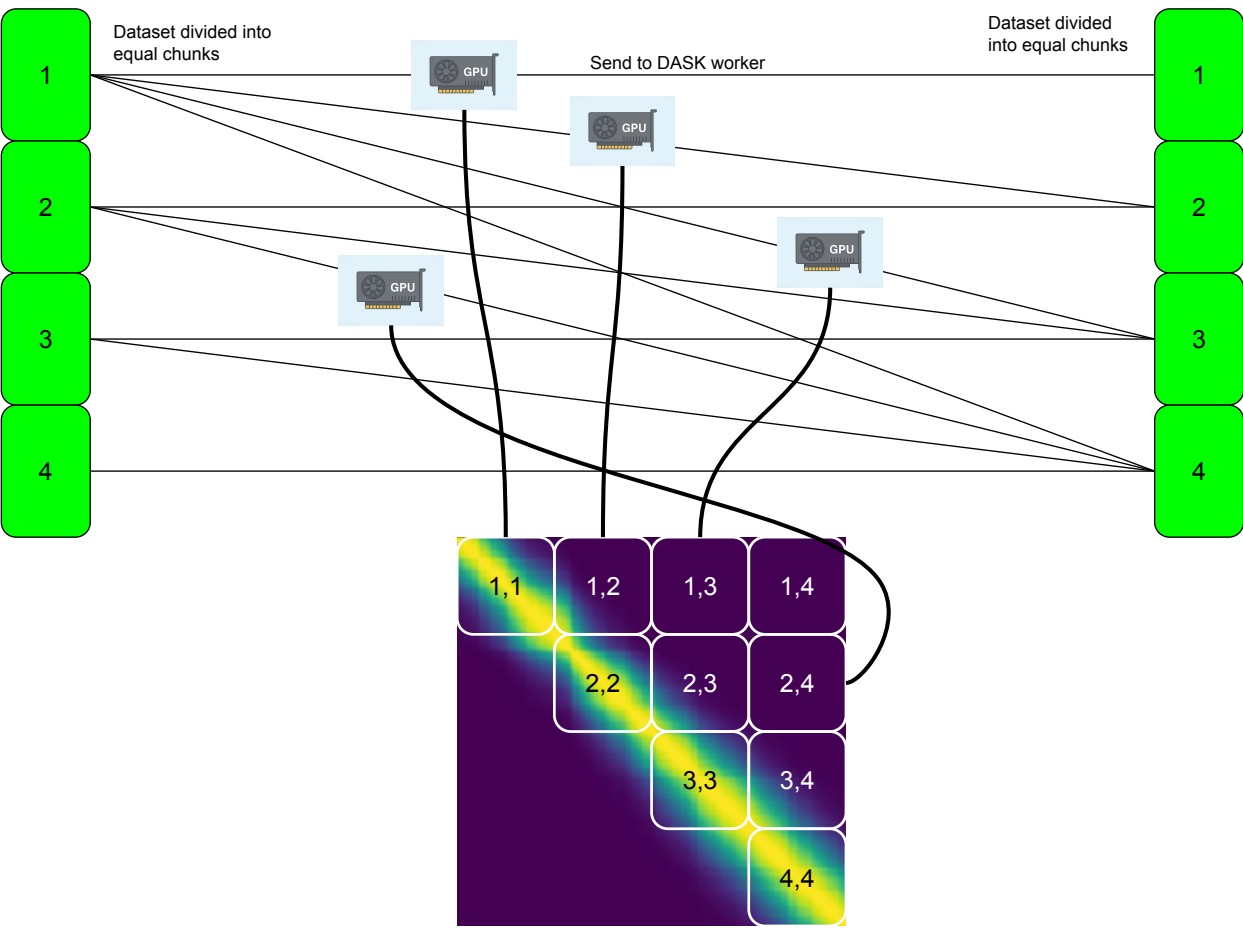

*Figure 2.* Compute pipeline. The dataset is divided into approximately equal chunks. Those chunks are sent to (DASK) compute workers, which calculate a dense square block of the covariance matrix. The block will be cast to sparse COO format before being shipped back to the host, where all blocks will be collected and assembled into the full sparse covariance matrix. This simple pipeline allows us to calculate truly massive covariance matrices in a reasonable amount of time. The kernels presented in this paper allow the discovery of sparsity in the covariance matrix, enabling downstream operations to be comparably cheap.

where $k_{\mathcal{M}}$ is any kernel of the Matérn class. Combining this kernel as shown in Equation 9 will lead to a vanishingly small influence of the bump functions. Instead

$$k(x_i, x_j) = \frac{1}{2}\sigma(x_i)\sigma(x_j)$$

$$\Big( \frac{|\mathbf{\Sigma}(x_i)|^{1/4}\,|\mathbf{\Sigma}(x_j)|^{1/4}}{\left|\frac{\mathbf{\Sigma}(x_i)+\mathbf{\Sigma}(x_j)}{2}\right|^{1/2}} k_{\mathcal{W}}(\sqrt{Q(x_i,x_j)})+$$

$$\frac{|\mathbf{\Phi}(x_i)|^{1/4}\,|\mathbf{\Phi}(x_j)|^{1/4}}{\left|\frac{\mathbf{\Phi}(x_i)+\mathbf{\Phi}(x_j)}{2}\right|^{1/2}} k_{\mathcal{M}}(\sqrt{P(x_i,x_j)})\sum_a g^a(x_i)g^a(x_j)\Big),$$

$$(12)$$

where $P$ is the equivalent of $Q$ but with potentially different hyperparameters, and $\mathbf{\Phi}$ is equivalent to $\mathbf{\Sigma}$ defined in Equation (6), will allow both terms to stay influential.

## C.3. Strategies for Kernel Design

Choosing a kernel can be a daunting task for standard GPs, and this is only aggravated for non-stationary compactly supported kernels proposed here. We want to provide some recommendations for that choice. For advanced kernel designs, it is often a good idea to start simple and move to a more complex kernel step by step, while observing changes in sparsity, log marginal likelihood, and prediction performance as the change unfolds. That means, we often start with a stationary anisotropic Wendland kernel (anisotropic version of (3)), and then add increasingly expressive non-stationarity in kernel (6). While the bump-function extension seems appealing, it is only worth considering when far-field interactions are suspected. When in doubt, this can be tested by allowing a few bumps, limiting the number of extra hyperparameters. If an improvement in log marginal likelihood is observed, more bump functions can be added. The MCMC prior can be used to automatically regularize the kernel and remove unnecessary bump functions. For discrete input spaces, such as the MNIST dataset, collapsing the bumps into deltas at the inputs can yield a more efficient mask, since the analytical shape of the bumps may impose a correlation structure that is inappropriate for discrete inputs rather than one discovered via optimality or sampling. The product kernel we defined, $k_{\mathcal{W}}k_d$, yields greater numerical robustness and improved approximations due to the smoothness imposed by the Wendland kernel. In short, starting simple — a stationary Wendland — and increasing complexity towards a full, non-stationary bump-based model is recommended and must be done cautiously while monitoring performance metrics. We want to demonstrate this strategy for each experiment

- **Exp. 1, 1d Synthetic Function:** The input space is a one-dimensional Euclidean space, and the latent function is smooth. We suspect no far-field interaction in this case, so we exclude any bump functions from the kernel. However, the test was set up with strong non-stationarity built in. This leads us to choosing kernel (6). That kernel is so diverse that we expect a majority of GPs could use it. It is certainly a good starting point for many situations in Euclidean settings.

- **Exp. 2, Topography:** The input space is a two-dimensional Euclidean space. The latent function is assumed to be smooth. Topography comprises strong non-stationarity and non-proximity-related far-field interactions that can play a substantial role. This led us to testing two kernels (6) and (9).

- **Exp. 3, CA Housing:** The housing dataset is defined on an eight-dimensional Euclidean space. The top priority was to keep the number of hyperparameters low. Scarcity in the dataset makes it extremely difficult to identify bump-function locations. In this scenario, simplicity is the primary objective, which leads us to use the kernel (3).

- **Exp 4, MNIST:** The MNIST dataset is defined on a set of handwritten digits. We defined the $l_1$ norm as a distance measure. Yet again, in such an abstract and high-dimensional space, simplicity and sparsity are key. We targeted a flexible and efficient mask via the collapsed bumps (deltas), while ensuring smoothness, a favorable spectral density, and stable computation through the product with the Wendland kernel.

- **Exp 5, 3d Climate** This dataset is defined on a three-dimensional Euclidean dataset. Because of the size and the associated covariance computation time, we wanted to keep the number of MCMC iterations to a minimum, which led us to keep the number of hyperparameters small. Yet again, simplicity was key. However, this is a prime example in which, in production mode, a more complex kernel could have improved overall approximation performance. This is the very example where future work will continue with a fully equipped bump-function kernel and additional computing resources.

## C.4. Intuition on the Number of Bump Functions

Kernel (9) has a practical shortcoming: the choice of the number of terms in the sums. More specifically, the far-field kernel $\sum_u^U g_u(x_i)g_u(x_j)$, where $g_u(x) = \sum_p^P b_u(x, x_p)$ needs the specification of $U$ and $P$. For intuition, imagine dividing the dataset into many subsets. Now, one might group the subsets by the covariances of their data; subset pairs with large cross-covariances are grouped together. $P$ can now be interpreted as the number of subsets in each group, and $U$ as the total number of those groups. $U = 1$ leads to a rank-1 far-field term, which means data-subsets

that are within support will see the same covariance contribution. $U = 2$ creates small groups of data subsets that co-vary similarly. If a third area covaries, the covariance structure can be achieved by summing over $u$. Furthermore, $P$ controls sparsity: as $P$ approaches $|\mathcal{D}|$, the size of the dataset, sparsity disappears. $U$ controls the rank of the far-field Gram matrix: if we need to approximate complicated functions or many orthogonal modes, we have to increase $P$. We want to remind the reader that GP training should start with all bumps disabled; they will be enabled only when a beneficial impact on the log marginal likelihood is detected.

## D. Training via Block-MCMC

The performance of the proposed algorithm is highly dependent on the setup of the block-MCMC. Our strategy is as follows. Hyperparameters of the kernel involving bumps are excluded from the block that samples the Wendland or core kernel hyperparameters. Within that block, we usually separate the amplitudes, radii, and positions. There is a trade-off to consider: a richer blocking strategy yields a faster stationary posterior; however, each additional block requires an additional log-likelihood evaluation per MCMC iteration. This tightly connects the MCMC to the kernel design and supports what we proposed there: starting with a simpler kernel and moving to a more complex one only when needed to increase sparsity and performance. The prior selection deserved another remark: it is acceptable to simply define a flat prior within well-chosen hyperparameter bounds; however, a well-chosen log-normal or Gamma prior over length scales or radii of the Wendland kernel and bump functions can improve sparsity substantially while leaving the final log-likelihood unaffected. Proposal distributions are kept at normal throughout our experiments.

## E. Visual Inspection of the *gp2Scale* Solution for the Topography Dataset

To offer the reader a more comprehensive way to visually study the posterior that *gp2Scale* produces, beyond our one-dimensional example, we provide a plot of the predictions and uncertainty estimates on a finely sampled grid.

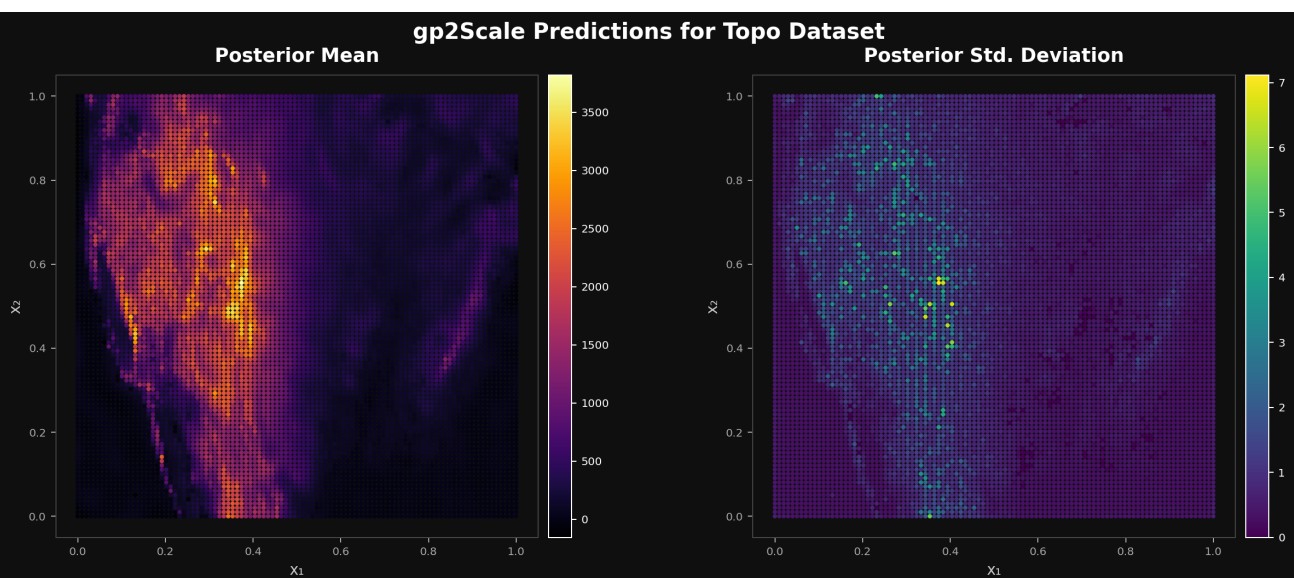

*Figure 4.* Posterior predictive distribution of *gp2Scale* on the U.S. topography dataset. Posterior mean (left) and posterior standard deviation (right) evaluated on a 100 by 100 prediction grid using kernel (6). The mean closely reproduces the large-scale elevation gradients of the Rocky Mountains and Appalachians while simultaneously resolving fine-scale local relief — a direct consequence of the non-stationary, spatially adaptive length scales. The posterior standard deviation reflects the inductive bias of the kernel: uncertainty is elevated in data-sparse interior regions and near sharp topographic transitions, where the compactly supported kernel assigns a lower effective correlation range, and is suppressed in the densely sampled coastal plains. This structured, spatially heterogeneous uncertainty pattern — rather than a uniform confidence band — demonstrates that the kernel class induces a function prior that is sensitive to local function structure, data density, and smoothness, consistent with well-calibrated uncertainty quantification on a strongly non-stationary real-world surface.

