# OpenReview forum: "gp2Scale: A Class of Compactly Supported Non-Stationary Kernels and Distributed Computing for Exact Gaussian Processes on 10 Million Data Points"
_ICML.cc/2026/Conference — ICML 2026 regular_

### Official Review · Reviewer_eXJ4 · 2026-03-11

**Soundness:** 2
**Presentation:** 2
**Significance:** 3
**Originality:** 3
**Overall Recommendation:** 4
**Confidence:** 4

**Summary:**

The paper is based on the premise that the choice of the kernel in Gaussian process regression can help unveil natural sparsity in datasets
The authors use Wendland kernels to ensure positive-definite sparse covariance structure and bump functions to modulate local support, yielding non-stationary kernels that adaptively induce sparsity in the data. They are basically interpreting the GP covariance with a graph-Laplacian like structure. As for inference, they target exact inference by leveraging the sparse structure of the covariance matrix to compute the log determinant and the inverse. The covariance matrix is constructed in parallel across GPUs using blockwise kernel evaluations, transmitted in a sparse format and assembled as a global CSR (compressed sparse row) matrix.

**Compliance With Llm Reviewing Policy:**

Affirmed.

**Final Justification:**

The rebuttal has addressed some of my questions and clarified some concerns, in light of which I have chosen to raise my score to a 4.

**Key Questions For Authors:**

1. The final kernel proposed in the paper has the structure of a core kernel (domain-specific) multiplied with the non-stationary component which modulates compact support, so the compact support kernel acts as a gating function controlling the interaction radius. Could the authors clarify how this form preserves the ability of the core kernel to model far-field interactions (as they say), given that the compact support enforces strict locality?

2. If each MCMC iteration costs ~8 minutes (this is for the 1mn subset of the temperature data), why not simply optimize the marginal likelihood instead? I didn't appreciate why the fully Bayesian inference scheme is warranted here. Most of the GP literature uses ML-II, if the paper was about proposing a new inference scheme I would understand the need for fully Bayesian but in larger datasets the benefits of fully Bayesian are more subdued.

**Limitations:**

A better discussion of limitations is warranted. In particular, outlining the conditions under which sparsity may deteriorate or inference may become computationally demanding would help clarify the practical applicability of the approach.

**Strengths And Weaknesses:**

Strengths:

- Significance: The concept of encoding algortihmic sparsity through the kernel design is an underrated idea and valuable to the GP community, which for the longest time has focused on inducing point based approaches and abstracted away the problem of kernel choice.
- Presentation is ok but can be improved to give more insight into the non-stationary kernels and the sparsity inductive bias.
- The work brings to the fore known kernels and combines them to construct non-stationary kernels that induce sparse covariance structures, it sparks ideas about how practitioners can think about kernel design as a template.

Weaknesses:

- Very limited discussion about the inductive biases these kernels induce on a function class, may be worthwhile to demonstrate these in a 2d setting. it is useful to visualise how the functions look for some fixed values of the hyperparameters.
- Computational cost of hyperparameter inference. The authors report a runtime of 477 seconds per MCMC iteration for a 3d 1mn dataset, since practical MCMC inference would typically require many such iterations, authors need to provide more practical alternatives (otherwise, it undermines the whole scalability story of the paper)
- Experimental evaluation could better isolate the contribution of the proposed kernel design. It is not always clear whether the competing methods are evaluated with comparable kernel choices. Since kernel expressivity can significantly influence performance, I am not sure if the authors are proposing a apples-to-apples comparison.

---

> ### Author Rebuttal · Authors · 2026-03-27
>
> We thank the reviewer for their insights, suggestions, and questions. We will address them all one-by-one. If the reviewer agrees with all or some of our explanations and finds value in the added content, please consider increasing the score. We would truly appreciate that. Thank you.
>
> 1. Inductive biases: A figure showing the posterior mean and variance is a great suggestion, and we added one to the appendix.
>
> 2. Cost of hyperparameter inference: This is a good point and a concern. We choose MCMC to propagate hyperparameter uncertainties into the posterior, which has been shown to be highly important for non-stationary kernels such as those we are proposing. It is usually true that the MCMC would traditionally need many iterations to become stationary. This is an important point. Our hyperparameters have physical meaning and can often be initialized to realistic values. In addition, careful blocking of the MCMC leads to much faster convergence, often in about 1000-5000 iterations. We added an explanation of this blocking strategy in the Appendix. Although we have considered ML-II as a training tool, it has not yet paid off. The number of function evaluations is comparable to our MCMC; UQ is not as well-calibrated. Local optimization cannot be considered due to costs. However, we appreciate the discussion, and this gives us new motivation to investigate alternative training strategies.
>
> 3. Apples-to-apples comparison: Although it is true that more complex kernels can be used in competitor methods, it was important to us to show baselines that are most useful for the reader. For all methods we compare to, extensions exist, but often lead to complicated rabbit holes where theory is scarce. For example, any non-stationary kernel can be incorporated into Vecchia, but there is no consensus in the community on how to best order the data and choose neighbors in this case.  Strong non-stationarity and any neighborhood approach (Vecchian, VNNGP) are a difficult mix. Our tests establish baselines with regard to accepted, well-known approaches in the community. They are not meant to present a comparison to fully customized and optimized implementations of the other methods. We added a sentence in section 5 to make this clearer to the reader.
>
> 4. Guide to kernel selection (points 2 and 3 in `strengths`): We added to the Appendix 3C guide for each experiment, explaining how kernels were chosen to make it easier for readers to apply the proposed method
>
> We also want to answer the reviewer’s questions one-by-one:
>
> 1. The simple product form using k_W alone does mute interactions outside the support, and the manuscript states this explicitly. This is why Sec 4.3 introduces the bump-function construction – Eq. (9), which is the first kernel in the class that can flexibly model far-field interactions. Sec 4.5 then addresses the specific failure mode when small length scales in k_core suppress the far-field term; it does this by separating the far-field components into k_core1*k_W + k_core2*\sum_p{g_p(x_i)*g_p(x_j).
>
> 2. The gp2Scale framework does not strictly require fully Bayesian training. As stated in Sec. 3, training can be improved by sampling from, or by maximizing the log ML. We use block MCMC because the proposed kernel family is expressive, thus yielding a strongly non-convex objective with many hyperparameters, and parameter uncertainty has to be propagated into the posterior, which is something only the fully Bayesian formulation does. That said, the scalability contribution is driven by the kernel-induced sparse covariance structure and the distributed COO→CSR assembly pipeline. It is not driven by MCMC.
>
> Further, we do not claim that gp2Scale dominates on raw wall-clock or hardware efficiency. We humbly claim that when exactness and kernel/noise customizations matter, then the method makes exact GP training feasible at scales previously well out of reach. In any case, we agree that there is room for improvement in optimizing the wall-clock time, and defer such intricacies to future work.
>
> Changes to the manuscript in response:
> - Sentence in section 5 regarding baselines and kernels
> - New figure visually presenting the topography result in Appendix E. It shows posterior mean and covariance side-by-side, and the caption reads, in part (due to character constraints) "Posterior predictive distribution of \emph{gp2Scale} on the U.S. topography dataset. Posterior mean (left) and posterior standard deviation (right) evaluated on a 100 by 100 prediction grid using kernel \eqref{eq:wendconv}. The mean closely reproduces the large-scale elevation gradients of the Rocky Mountains and Appalachians while simultaneously resolving fine-scale local relief --- a direct consequence of the non-stationary, spatially adaptive length scales. ..."
> - Explanation of the MCMC training in the appendix
> - Guide to kernel selection in the appendix.

---

> > ### Author Rebuttal · Reviewer_eXJ4 · 2026-04-05
> >
> > Thank you for the response, I am happy to raise my score in light of my better understanding of the contributions of the paper.
> > However, the main unresolved point for me is the inference choice, I understand the motivations of using MCMC to sample the hypers; however, the authors claim that ML-II is an incompatible inference choice. The statement below:
> >
> > "Although we have considered ML-II as a training tool, it has not yet paid off. The number of function evaluations is comparable to our MCMC; UQ is not as well-calibrated. Local optimization cannot be considered due to costs"
> >
> > what is meant by local optimisation. What about the baselines, were they not trained with ML-II?

---

> > > ### Author Response · Authors · 2026-04-06
> > >
> > > We are very happy about the positive interaction and the raised score. Thank you. We want to answer the remaining question here, and we apologize for not being clearer in our previous answers.
> > >
> > > Clarifications on the baselines and training for method comparisons:
> > > All competitor methods come with built-in recommended training routines. SVGP, for example, is based on variational inference, which typically involves optimizing the ELBO using a local optimization algorithm, here ADAM. This was the recommended setting for the algorithm we compared to. We want to compare our methods with competitors in their best, most well-known, and widely accepted forms. Therefore, we went with those settings for all our competitors. Every GP we implemented (gp2Scale and the base GP) was trained with MCMC.
> > >
> > > Why is MCMC ideal for us but not for other methods?
> > > One of our method's strengths is the complete agnosticism to kernel design. Non-stationary kernels with many hyperparameters are difficult to train, and uncertainty in the int eh hyperparameters matters for the final posterior. MLEII and local optimizers would not lead (without any post-hoc tricks) to the inclusion of hyperparameter uncertainty. Methods that use simpler kernels (simply because they would not allow or cooperate well with non-stationary kernels) are less sensitive to hyperparameter uncertainty and can therefore be trained using optimization (MLEII or local, mostly ADAM). Too many latent variables mean sparse variational inference will perform better than an MCMC could (SVGP).
> > >
> > > In summary, we wanted to show the reader comparisons with competitor methods at their best baseline, that is, kernels that are well established and training methodologies tailored to their unique structure.

---

### Official Review · Reviewer_eGKQ · 2026-03-12

**Soundness:** 2
**Presentation:** 3
**Significance:** 2
**Originality:** 2
**Overall Recommendation:** 4
**Confidence:** 4

**Summary:**

This work presents the idea that exact Gaussian Processes are possible at scale (1e7 points were tested) if enough computational resources are devoted, if there is natural sparsity in the data and if the kernel is endowed with the ability to discover it so that sparse matrix algorithms are possible.

**Compliance With Llm Reviewing Policy:**

Affirmed.

**Final Justification:**

I believe I am better understanding the authors' perspective, which is that the benefit of maintaining the exact GP with scale is being able to use complex non-stationary kernels without too much fuss other than needing MCMC training because MLE fails (as has been revealed in the rebuttal phase). But it would be worth knowing if I can just use another baseline methods with even a slightly more expressive kernel (such as RQ kernel, anisotropic kernel, or SKI with its native Spectral kernel) instead of needing this approach. Indeed it is remains hard to compare GP methods where one has a very highly parameterized kernel and one has the absolute bare-minimum kernel. My concern with conceptual novelty compared to Noack et al. 2023 also remains.

Given the authors' candidacy about limitations, contributions, and their responsiveness, addressing a good number of the other reviewer's concerns, I will raise my score to a 4.

**Key Questions For Authors:**

See weaknesses

**Limitations:**

yes

**Strengths And Weaknesses:**

Strengths:

1) The work is well written and effectively places itself in the context of existing methods.
2) The work is amenable to customization for users to plug in their own kernels (although this facet is not thoroughly tested).
3) Authors train their GP on a 3d dataset with 10 million points, which is an impressive feat.

Weaknesses:

1) Existing work (Noack et al. 2023) already considers scaling exact GPs via sparsity discovering kernels. The authors candidly acknowledge this, saying this manuscript adds more 'kernel designs', more 'comparisons to other scalable GP methodologies' and an 'accompanying software release.' But nonetheless the conceptual novelty is limited.

2) For baselines, the authors used a relatively non-expressive kernel (isotropic Matern 3/2) compared to theirs. While they at least tried and failed to use theirs in SVGP on the synthetic data (line 301), many other expressive kernels exist (e.g. Gibbs kernel, [1], [2]). For one, this makes it hard to quantify the expressivity of their kernel over existing expressive kernels. An ablation study over different kernels using their training method (to isolate the kernel's effect) would ameliorate this. Also, this raises concerns about the comparison being fair, for instance, the SKI method used a Gaussian Spectral Mixture [1] by default in its original implementation, which was not used by the authors in their baseline.

3) In the same vein, the authors used MCMC for their method and maximum likelihood training for baselines, which confounds results. The authors could train their exact GP with MML also to give the readers a sense of MCMC's effect.

4) Lack of plots to accompany empirical studies. A posterior predictive plot on a real world dataset would give the readers a better idea of fit and uncertainty calibration as only one is shown for the synthetic data.

References:

[1] Gaussian Process Kernels for Pattern Discovery and Extrapolation, arxiv: https://arxiv.org/abs/1302.4245

[2] Non-Stationary Spectral Kernels, arxiv: https://arxiv.org/abs/1705.08736

---

> ### Author Rebuttal · Authors · 2026-03-27
>
> We thank the reviewer for their assessment of our work and the actionable suggestions. We want to organize our response point-by-point. In response to our changes, we would greatly appreciate a higher score. Thank you for your consideration.
>
> 1. Novelty: We appreciate the reviewer’s perspective here. The basic concept itself is not entirely novel, but that is true of much of current ML work, which is largely based on extensions of existing methodologies. In this case, the mentioned paper is a very early version of the core methodology, had no practical implications for users, and was untested. It focused mostly on computational aspects rather than on a useful kernel framework. This work makes clear contributions that are outlined in the paper. It introduces entirely new kernel formulations, proposes kernels for non-Euclidean settings (such as MNIST), and, for the first time, conducts rigorous comparisons and provides an open source software framework. We argue, therefore, that the novelty is sufficient for publication.
>
> 2. Expressive Kenrels: Although it is true that more complex kernels can be used in competitor methods, it was important to us to show baselines that are most useful for the reader. For all methods we compare to, extensions exist, but often lead to complicated rabbit holes where theory is scarce. For example, any non-stationary kernel can be incorporated into Vecchia, but there is no consensus in the community on how to best order the data and choose neighbors in this case.  Strong non-stationarity and any neighborhood approach (Vecchia, VNNGP) are a difficult mix. For any method that includes variational inference, complex non-stationarity leads to errors in the posterior that are difficult to quantify. Our tests establish baselines with regard to accepted, well-known approaches in the community. They are not meant to present a comparison to fully customized and optimized implementations of the other methods. We added a sentence in section 5 to make this clearer to the reader.
>
> 3. Training: This is true and a very good point. However, in each case, we chose the most natural training strategy for the method at hand. SVGP, for instance, trains the inducing point positions via variational inference and ADAM; any other training strategy would be unnatural for the algorithm and invite a host of issues. In our Vecchia code, MLE was the natural choice because there are only a few hyperparameters, and the code used operates best in this setup. Often, the latest software tools come with the training strategy baked in (at least by default and for good reason), and changing it could create a host of issues with ample opportunity for choices to be made, leading to more ambiguity in the tests. We believe it is most valuable for the reader to present a fair baseline for the algorithms, as they were designed to perform optimally. Our method, like many that use non-stationary kernels, performs well for the fully-Bayesian MCMC because hyperparameter uncertainty is propagated into the posterior. However, we are actively testing new training paradigms and will reconsider MLE in the future.
>
> 4. Empirical Results: Thank you for this great suggestion. We added a plot for the topography dataset in the appendix showing predictions and posterior standard deviations. This is a great addition. Thank you for this.
>
> Changes to the manuscript:
> - a sentence explaining the competitor baselines in section 5
> - added explanation of the MCMC approach in the appendix
> - Figure showing empirical results for the topography dataset (posterior mean and variances) (appendix E) side by side. The caption reads: "Posterior predictive distribution of \emph{gp2Scale} on the U.S. topography dataset. Posterior mean (left) and posterior standard deviation (right) evaluated on a 100 by 100 prediction grid using kernel \eqref{eq:wendconv}. The mean closely reproduces the large-scale elevation gradients of the Rocky Mountains and Appalachians while simultaneously resolving fine-scale local relief --- a direct consequence of the non-stationary, spatially adaptive length scales. The posterior standard deviation reflects the inductive bias of the kernel: uncertainty is elevated in data-sparse interior regions and near sharp topographic transitions, where the compactly supported kernel assigns a lower effective correlation range, and is suppressed in the densely sampled coastal plains. This structured, spatially heterogeneous uncertainty pattern --- rather than a uniform confidence band --- demonstrates that the kernel class induces a function prior that is sensitive to local function structure, data density, and smoothness, consistent with well-calibrated uncertainty quantification on a strongly non-stationary real-world surface."

---

> > ### Author Rebuttal · Reviewer_eGKQ · 2026-04-03
> >
> > I believe the baseline comparison remains unfair regarding the kernel choices. If non-stationary kernels pose issue as the authors mention, there are expressive kernels which are stationary, i.e. the spectral kernel supposed to be used by default in SKI.
> >
> > In general, it is hard to decouple the effect of the proposed kernels versus the effect of the GP being exact. All the methods would have no problem training on the vanilla Wendland kernel correct? This would 1) help isolate the effect of the GP being exact compared to the other scalable methods and 2) it would show how much the complex nonstationary kernels contribute to performance of the authors' method compared to just a one-scale parameter kernel.
> >
> > It also remains hard to understand the decision to use MCMC training given MML's ubiquity. For instance, if the exact GP with this highly parameterized kernel just fails to train well under MML, this is an illuminating point and should be made clear to the reader.
> >
> > In line 529 of the manuscript: "Vecchia and gp2Scale are trained via MCMC," but here authors mention they train Vecchia with MLE. Which is it?

---

> > > ### Author Response · Authors · 2026-04-03
> > >
> > > Dear Reviewer, we try to address these remaining concerns one-by-one:
> > > "I believe the baseline comparison remains unfair regarding the kernel choices. If non-stationary kernels pose issue as the authors mention, there are expressive kernels which are stationary, i.e. the spectral kernel supposed to be used by default in SKI."
> > >
> > > We want to reiterate our belief that a baseline comparison is more useful to the reader than a comparison of a kernel we customize by hand without being experts in that method. We can look at a couple fo example, starting with SKI. Yes, absolutely, extensions for flexible kernels exist in SKI, but the spectral kernel potentially comes with many hyperparameters, and that depends on the dimensionality of the problem. This means new bounds for the hyperparameters and sensitivity to training strategy. We would have to make many choices that the reader and reviewer may or may not agree with, increasing ambiguity in the tests.  For Vecchia, the same is true. Let's say we implement the same non-stationary kernel for Vecchia. A careful reviewer would immediately criticize the lack of a clear sorting and neighborhood selection process.  For SVGP, variational inference is used, and it is deeply coupled with ADAM for several reasons. ADAM will not perform well for non-stationary kernels because of many local optima. So yes, we can basically implement the same non-stationary kernel, but it would lead to tests that are not valuable because of the arbitrary choices we would be forced to make. We stayed on the safe side with comparisons to competitor baselines. Nowhere do we claim that our method will always, for all datasets and customizations, outperform all other methods, and we are very honest about this in the manuscript.
> > >
> > >
> > > "In general, it is hard to decouple the effect of the proposed kernels versus the effect of the GP being exact. All the methods would have no problem training on the vanilla Wendland kernel correct? This would 1) help isolate the effect of the GP being exact compared to the other scalable methods and 2) it would show how much the complex nonstationary kernels contribute to the performance of the authors' method compared to just a one-scale parameter kernel."
> > >
> > > The main reason we avoid this isolation is that the inherent stationarity is exactly the strength of the proposed method.
> > > The agnosticism about non-stationarity is exactly what we want to exploit; many competing methods are not natively capable of leveraging it. In other words, we don't want to make the claim that given the same kernel, exact GPs are more accurate. Of course they are in many cases, and this is well known. We make the point that our scalable method is, by design, non-stationary, and therefore, we can take advantage of that.
> > >
> > >
> > > "It also remains hard to understand the decision to use MCMC training, given MML's ubiquity. For instance, if the exact GP with this highly parameterized kernel just fails to train well under MML, this is an illuminating point and should be made clear to the reader."
> > > We included an additional explanation of why maximum-likelihood training is suboptimal for our method. We appreciate the suggestion.
> > >
> > >
> > > "In line 529 of the manuscript: "Vecchia and gp2Scale are trained via MCMC," but here authors mention they train Vecchia with MLE. Which is it?"
> > > Thank you, we fixed this discrepancy in the manuscript.
> > >
> > > In summary, while the conversation about the exact way how tests are conducted is very valuable --- and we appreciate the reviewer's insights --- it does not lower the value of the results that have been obtained. Given that and the fact that we did improve the manuscript and fixed some of the mentioned issues, we would very much appreciate a "weak accept" (4). Thank you for your consideration.

---

### Official Review · Reviewer_uNNc · 2026-03-12

**Soundness:** 3
**Presentation:** 2
**Significance:** 3
**Originality:** 4
**Overall Recommendation:** 5
**Confidence:** 4

**Summary:**

This paper discusses a strategy to scale up Gaussian process (GP) modeling for a large number of points, an alternative to the existing, various approximations based on inducing points or neighbors. The method for doing so is to design a kernel different from the usual stationary ones, a nonstationary kernel with compact support which permits sparse covariance matrix computation. As the authors mention, while the use of compactly supported kernels is not new in the stationary context ("covariance tapering"), non-stationarity helps mitigating the resulting potential inflexibility in modeling long-range dependencies. Concrete proposals for designing such a kernel is discussed in Section 4.

**Compliance With Llm Reviewing Policy:**

Affirmed.

**Final Justification:**

My comments on minor typographical errors have been addressed, and I have no further follow-up questions regarding the substance of the work. The work in my view maintains interesting ideas about kernel designs and clear exposition of the requisite compute pipeline. I thus suggest its acceptance to ICML.

**Key Questions For Authors:**

1. I wonder if one of the reasons that indudcing point-based or Vecchia approximation is hard to beat is because they implicitly are fitting a kernel with much more complicated non-stationary structure than the stationary kernels they purportedly approximate. There has been some recent theoretical studies on the Vecchia GPs viewed in their own right and not in terms of the kernels they approximate; I wonder if you have considered this line of inquiry.
2. While you have described in detail the sparse computation pipeline for your proposed kernel, I think a more natural way to reap benefits is by developing a kernel whose induced __precision matrix__ is sparse, as done previously in the context of spatial GP models on manifolds. Do you have ideas about such a precision-based design, and are some of your ideas in this work also useful in designing a sparse precision matrix?

**Limitations:**

Yes.

**Strengths And Weaknesses:**

The work is an important contribution in my opinion, both because the proposed method allows scaling up Gaussian process inference to much larger datasets than previously envisioned, and also because, viewed in its own right, the design principle of a covariance kernel that is both non-stationary and compactly supported is interesting in its own right. It is also to the authors' credit that they make clear the limitations and trade-offs involved in choosing their proposed kernel design, when honestly compared with alternatives (e.g., inducing points or Vecchia approximation to stationary GPs).
Some mathematical exposition of the paper may need improvements. For instance, in Section 4, the definition and notation for the bump function is confusing: after eq. (8), the authors assert $g(x_i)g(x_j) = \sum_{pq}^{P} b(x,x_{p}) b(x,x_{q})$, and I am not quite sure if the bound variables $x$ should read as $x_{i}$ and $x_{j}$. Equations (3), (6) and (7) also needs to be re-formatted.

---

> ### Author Rebuttal · Authors · 2026-03-27
>
> We sincerely thank the reviewer for their positive assessment of our work. We fixed all the above-mentioned notation and formatting issues in the new version of the manuscript. All equations and nomenclature have been checked regarding the mentioned issue. We genuinely appreciate the in-depth look the reviewer took to provide valuable comments. We are very grateful for the score we received, but would of course very much appreciate an increased score in response to the improvements we made.
>
> We also want to offer answers to the raised questions:
> 1. This is a very insightful question and demonstrates the reviewer’s expertise in the field. It is, in our opinion, true that both approximate methods are implicitly inducing non-stationarity because of the chosen inducing-point/neighbor choice. However, whether this choice is amenable to optimal hyperparameter training is less clear. Neighbor selection and ordering, for Vecchia, are far from straightforward in a truly non-stationary setting. The same is true for the placement of inducing points. In addition, variational inference for the placement of inducing points might introduce additional errors that are difficult to quantify.
>
> 2. We are thinking along similar lines. Yes, we have considered precision kernels. However, because of the spectral properties of all common kernels, they do not lead to desirable smoothness properties of the approximated function. Our bump function kernel with collapsed radii could theoretically serve as a precision kernel, but it would be very difficult to train. Coming up with general precision kernels with the right spectral properties yields differential operators for SPDEs whose solutions are Matern kernels and, when solved numerically, yield a sparse precision matrix; however, this only works on meshes and, therefore, relatively low-dimensional Euclidean spaces.

---

> > ### Author Rebuttal · Reviewer_uNNc · 2026-04-02
> >
> > I thank the authors for their response. My comments on minor typographical errors have been addressed, and I have no further follow-up questions regarding the substance of the work. I am retaining my original score as it is, which is to accept the paper.

---

> > > ### Author Response · Authors · 2026-04-02
> > >
> > > We are very thankful for the constructive comments on the original manuscript and for the acknowledgment of the rebuttal.  We appreciate the score that was given. Thank you.

---

### Official Review · Reviewer_8QPT · 2026-03-13

**Soundness:** 4
**Presentation:** 3
**Significance:** 4
**Originality:** 4
**Overall Recommendation:** 5
**Confidence:** 2

**Summary:**

The paper proposes four non-stationary, compactly supported kernel formulations. These formulations enable sparsity to be discovered, thereby enabling a distributed computation framework and making it possible to compute an exact Gaussian process on large datasets.

**Compliance With Llm Reviewing Policy:**

Affirmed.

**Final Justification:**

After the rebuttal, I am increasing my score to 5. In the rebuttal, the authors addressed my main concern about the lack of computational details and intuition behind the kernel design by adding new Sections in the appendix that provide the requested details. In my opinion, this paper makes an important contribution to the scalability of GPs. Therefore, I recommend acceptance, with the expectation that the new sections will be included in the camera-ready version.

**Key Questions For Authors:**

- How are the kernels selected in each experiment? I think the paper would benefit from some intuition on when to choose each kernel. For example, why was the product of the kernel used in MNIST?
- Can the authors provide more details about the distributed computing and Block MCMC? The authors claim that one of the components of \textit{gp2Scale} is "a customised block- Metropolis-Hastings Markov-chain Monte Carlo (BMH-MCMC) that allows quick convergence, natural regularisation, and user-friendliness", but this is barely discussed in the paper, even in the appendix the authors only provide a figure.
- On the previous point, are all the hyperparameters of the kernels learned through marginal likelihood? I'm asking because, for example, the radius in equation 3 will affect the compactness of the kernel and therefore could affect how the computation is distributed.

**Limitations:**

Yes

**Strengths And Weaknesses:**

The paper is well written and easy to follow. The authors present the different classes of kernels in a natural way. In my opinion, the method has the potential to improve the computation of Gaussian processes and make it more user-friendly. Overall, I think it is an interesting idea and solution for a big problem in the GP community.

I would say the main weakness of the paper is the lack of detail in the computation. One of the key claims of the proposed method is the distributed computing for exact Gaussian Processes, but this is barely discussed in the paper, and mainly focuses on the class of kernels. Along those lines, I think the paper also lacks motivation/intuition for when each class of kernel should be preferred.

---

> ### Author Rebuttal · Authors · 2026-03-27
>
> We thank the reviewer for their overall positive assessment of our work and for the constructive suggestions to improve the manuscript. In response, we have added a more detailed description of the distributed computing framework we have developed. Please see Appendix B7. Among other content, we now describe in detail:
> - How the dataset is split up and distributed across nodes/cores or GPUs
> - How the kernel is called and what is communicated back to the host node
> - The matrix assembly process
>
> In addition, we added detailed information to build intuition about the kernel choice in Appendix 3C and a discussion of the block-MCMC setup in Appendix D. We sincerely hope this addresses all the reviewers’ concerns and that the reviewers will award us a higher score. We would very much appreciate this consideration.
>
> Answers to the questions:
> 1. The kernels are selected based on some experience and often, using plenty of caution. For example, in spatial statistics applications, local interactions play an important role and are highly non-stationary and anisotropic. This is where kernel (6) excels. If far-field interactions are suspected, bumps should be introduced slowly. For the MNIST dataset, a multiplicative kernel seemed reasonable, promoting high, highly flexible covariances but only locally within the support of the Wendland kernel. We added additional explanations to the manuscript to better guide the reader through the kernel selection process for each experiment.
>
> 2. We added detailed information to the manuscript regarding the block-MCMC setup. In short, bump-function hyperparameters are separated from the Wendland and core hyperparameters by blocks.  In addition, certain bump function hyperparameters are also separated. For instance, shape parameters and bump locations should not be in the same block.
>
> 3. Yes, the shape of the bump functions (including the radius) and the core kernel hyperparameters are all trained via sampling of the log marginal likelihood. We added more detailed information in the manuscript.
>
> Changes made to the manuscript, please see the new sections:
>
> \subsection{Distributed Computing Pipeline}\label{A:comp}
> The distributed-computing pipeline is shown in Figure \ref{fig:comp_pipe}. \rev{We provide a detailed, practical explanation here. The computational process starts by dividing the datasets into a number of subsets of a given size that depends on the distributed compute architecture. The distributed computing is handled via Python's DASK. The data subset size per worker depends on the compute resources each one has available and the particular kernel definition. The worker will use the kernel to calculate a block of the covariance matrix. Maximum RAM usage should not exceed the available RAM per worker and should optimally utilize the available compute. For example, the A100 GPUs we run many of our models on are optimally utilized at a block size of 15000 points. This number can often be smaller for CPU workers. Once the dataset is distributed to the workers, each worker calls the kernel function and computes the corresponding block of the covariance matrix. Still on the worker, the matrix block is cast into the sparse COO format to minimize communication burden between the worker and host nodes. Once all blocks are computed, the full sparse covariance matrix is assembled on the host node. Sparse linear algebra follows and may use the same workers. A user should ensure that there are at least as many blocks to compute as there are GPUs available, for optimal hardware utilization. When more blocks have to be computed than GPUs are available --- this is the most common case --- work will be optimally scheduled automatically. The object $K^{-1} \mathbf{y}$, where $K$ is the covariance matrix, is stored for fast posterior mean computation. The described process is repeated for each MCMC iteration for the training.}
>
> \subsection{Strategies for Kernel Design}
> Unfortunately, it is too long here, but it includes a detailed explanation of the reasoning behind each kernel.
>
> \section{Training via Block-MCMC}
> \rev{The performance of the proposed algorithm is highly dependent on the setup of the block-MCMC. Our strategy is as follows. Hyperparameters of the kernel involving bumps are excluded from the block that samples the Wendland or core kernel hyperparameters. Within that block, we usually separate the amplitudes, radii, and positions. There is a trade-off to consider: a richer blocking strategy yields a faster stationary posterior; however, each additional block requires an additional log-likelihood evaluation per MCMC iteration. This tightly connects the MCMC to the kernel design and supports what we proposed there: starting with a simpler kernel and moving to a more complex one only when needed to increase sparsity and performance. The prior selection deserved another remark: it is ... [truncated because of max character number reached]

---

> > ### Author Rebuttal · Reviewer_8QPT · 2026-04-01
> >
> > Thank you for the clarifications and the new Sections. I'm increasing my score accordingly.

---

> > > ### Author Response · Authors · 2026-04-02
> > >
> > > We very much appreciate the author's response and are happy that our clarifications were useful. We are excited about the increase in score! Thank you.

---

### Decision · Program_Chairs · 2026-04-30

**Decision:**

Accept (regular)

**Comment:**

This paper proposes a methodology to scale exact Gaussian processes to more than 10 million data points without relying on inducing points, kernel interpolation, or neighborhood-based approximations. Instead, the proposed method leverages kernel design. Highly flexible, compactly supported, and non-stationary kernels lead to the identification of naturally occurring sparse structure in the covariance matrix. This is then exploited for the calculations of the linear system solution and the log-determinant for training. The paper demonstrates the method's functionality on several real-world datasets and compares it with state-of-the-art approximation algorithms. The method shows superior approximation performance in many cases. The method is agnostic towards core kernel design, noise, and mean functions, and the type of input space. All the reviewers agree that this is a nice submission with clear advantages. The paper is well written and easy to follow. They think that this is an interesting idea and a solution for a big problem in the GP community. The work is amenable to customization for users to plug in their own kernels. Moreover, authors train their GP on a 3d dataset with 10 million points, which is an impressive feat. The reviewers have also indicated that the contribution is significant. Namely, the concept of encoding algorithmic sparsity through the kernel design is an underrated idea and valuable to the GP community. The reviewers have also identified some weaknesses of the submission. However, most of them have been properly addressed during the rebuttal. The final version of the paper should include the material explained during the rebuttal.